# Fleshing out the data: when epidemiological researchers engage with patients and carers. Learning lessons from a patient involvement activity

Melanie Morris,[1] Yuki Alencar [ID],[2] Bernard Rachet [ID],[2] Richard Stephens,[3] Michel P Coleman[1]

¹Cancer Survival Group, NCDE, London School of Hygiene & Tropical Medicine, London, UK
²Department of Non-Communicable Disease Epidemiology, London School of Hygiene & Tropical Medicine, London, UK
³Consumer Forum, National Cancer Research Institute, London, UK

**Correspondence to**
Yuki Alencar;
yuki.alencar@lshtm.ac.uk

## ABSTRACT

Patient and public involvement and engagement has become an essential element of health research, ensuring aims and outputs are worthwhile and relevant. However, research involving secondary data analyses does not present immediately obvious ways to involve patients and the public. Innovative approaches to ensure their involvement is meaningful and effective are required. The Cancer Survival Group cohosted a full-day meeting with the National Cancer Research Institute Consumer Forum—a group of patients and carers. This included the Forum's 'Dragons' Den': a small-group session in which their members provided insight, advice and ideas on current or planned research in the Cancer Survival Group. We investigated this activity as an example of effective patient involvement, with the aim of developing broad recommendations to improve epidemiological/quantitative research by involving patients and carers as directly as possible.

In addition to quantitative data captured through evaluation forms completed after the event, we used semistructured interviews of a sample of participants to evaluate the effectiveness of the session and to learn lessons. The interviews were analysed to identify broad or recurrent themes and recommendations.

Feedback was overwhelmingly positive, and some impacts on the research projects were identified. Interviewees commented on overall expectations and experiences, as well as specifics of room layout, timing of the session, composition of groups, effectiveness of the facilitation and content of discussions.

We present a summary of our findings as a guide for other researchers, including recommendations for improvement gleaned from the interviews. The value to researchers of hosting and participating in such activities was clear. We developed recommendations that should help to improve future events for ourselves and for others who wish to conduct similar activities, which in turn may lead to more concrete benefits for research and patients.

## INTRODUCTION

The Cancer Survival Group (CSG) at the London School of Hygiene & Tropical

## Strengths and limitations of this study

► Innovative approaches for involving patients and the public are needed for epidemiological research to ensure it is meaningful and effective.
► This novel format of involvement has been evaluated to provide insight and gather recommendations for improvements.
► Comments on the room layout, session timing, composition of groups and effectiveness of facilitation led to lessons learnt that will be implemented in the future.
► Findings can be used by other researchers in a variety of areas to establish effective forums for patient and public involvement (PPI).
► More needs to be done to ensure a broader range of patients and the public participate in PPI activities, to ensure the experiences and views of all sectors of society are included.

Medicine has actively involved patients and carers in its research for many years.[1] As members of the Group's Advisory Panels, they comment on research and propose ideas for further research or funding. In 2017, the CSG cohosted an event with the National Cancer Research Institute (NCRI) Consumer Forum with whom we have a well-established relationship. This event included members of the Forum contributing to our research through discussion of our projects, in 'Dragons' Den' style (from the well-known television programme), in which the patients or public are the 'dragons' to whom the researchers 'pitch' their research ideas. This and other forms of patient involvement are common in other areas of research,[2–5] but are not often attempted by epidemiological researchers.[6] Here, we set out the lessons

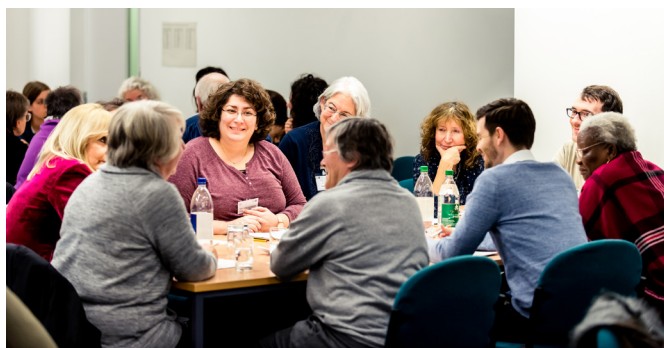

**Figure 1** In the 'Dragons' Den'. Credit: photograph by Simon Callaghan Photography.

learnt, to help other researchers capitalise on the involvement of patients.

### What is real patient involvement?

The involvement of patients, carers and the public (hereafter, for brevity, we will use the term 'patients' or 'patients with cancer' to refer to this group, which may include patients, carers and members of the public) in health research helps to ensure that the research being planned and the approaches undertaken to deliver it are relevant to their needs and interests.[7–10] It also provides insights from the experience of people who use health services, or live with the condition in question. INVOLVE, the UK's national advisory group for public involvement in research, defines it as 'research being carried out "with" or "by" members of the public rather than "to", "about" or "for" them'.[11 12] Patients and the public can offer new perspectives that researchers may be unable to provide, and can help improve the quality of the research.[7 10–12] They also highlight the importance of making health research more transparent and publicly accountable, as well as empowering patients themselves.[13]

As epidemiologists, we try not to forget that each data point represents a person, but we need to re-examine whether we are asking the right questions of the

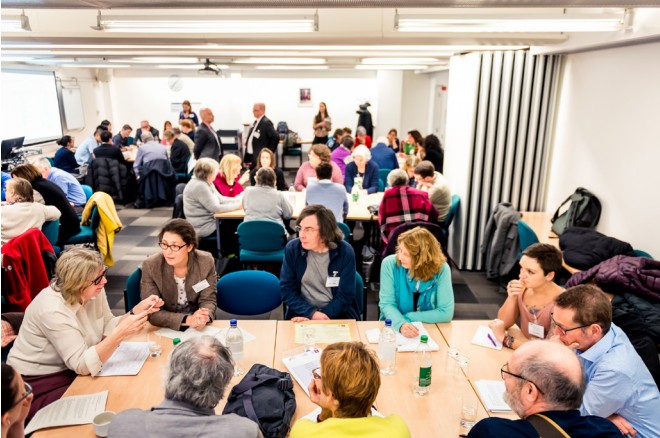

**Figure 2** The Dragons' Den session under way, showing the layout of the room used. Credit: photograph by Simon Callaghan Photography.

data—those that are most relevant to improving the health and well-being of patients. Researchers who carry out secondary analysis of routinely collected data, rather than from primary data collection such as patient surveys, may find it challenging to involve patients more effectively in their research.[14–16]

In this communication piece, we outline what we learnt from running a patient involvement activity. This was not intended to be a research project in itself, but we offer here some reflections on the experience that we hope will inform and benefit other researchers.

### AN EXAMPLE OF PATIENT INVOLVEMENT

The CSG cohosted a 1-day symposium with the NCRI Consumer Forum, for over 50 external invitees including patients with cancer and carers. Our aim was to showcase the Group's recent and current work to these guests, and to explain in detail how data are used in our work.

As part of the day, patients and researchers participated in an interactive session: the Dragons' Den. Closer in concept to breakout workshops than to focus groups, the idea of the NCRI Consumer Forum's Dragons' Den is to be a collaborative event in which patients are not just listening to presentations, but are equal participants in a round-table discussion; in which there is a shared desire to move research forward; and during which researchers can learn directly from patients' experiences and perspectives to shape and improve their research.[17]

Consumer Forum members had chosen one of five groups, each of which was led by a researcher in a discussion of a particular research question . These topics had been circulated in advance, with some of the questions the researchers wanted to explore (online supplemental appendix A). Following a brief presentation of the research area/question, the researchers asked the members for their input, advice and ideas around the planned research (figure 1). The session was followed by a short plenary session in which a patient representative from each table presented the main points discussed, allowing us to capture the salient points of conversations that had taken place at each table, and to identify any broader or recurrent themes that emerged on the day.

We describe the experience from this session to illustrate how researchers can interact effectively with patients and their representatives to improve research. We describe how the session was run, and the feedback and lessons we learnt. We offer recommendations for how future sessions could be planned, and how this technique can be used to make the most of opportunities for collaboration.

### Set-up

A large teaching room with maximum capacity of 70 persons was set up with five tables seating up to 10 people each (figure 2). On the day, 45 people attended, two-thirds (64%) of whom had been NCRI Consumer Forum members for 2 years or more. Most had expressed a preference for a particular topic. Those who had not were randomly assigned

to a table. Each table had a group of patients and two CSG researchers, one of whom acted as rapporteur. We scheduled 40 mins for the session, followed by a 45 mins plenary session for feedback. These timings were based on the usual duration for Dragons' Den sessions.

## Methods for evaluating the meeting

Forum members were initially sent feedback forms within a week of the meeting. The researchers were asked to provide notes and immediate reflections. We received both positive and negative comments (online supplemental appendix B). We have attempted to go beyond the usual 'polite' feedback to understand what was good about the session, but also what was challenging, so that we could identify lessons that other researchers could implement in their interactions with patients.

In the second phase, semistructured interviews with five patients and five researchers, one from each table, were carried out within 6 months of the meeting (online supplemental appendix C). These interviews were transcribed verbatim and thematically analysed, along with the notes collected immediately after the event.

The thematic analysis of the transcripts followed the stages of data familiarisation, coding and theme refinement. Transcripts were read, cleaned and read again by two authors (MM, YA). The texts were coded by individual researchers, who assigned a tentative theme to each meaningful phrase. Themes were then discussed, grouped together and recoded into overarching themes which are summarised below.[18 19]

## SUMMARY OF REFLECTIONS AND FEEDBACK

The following seven themes emerged from analysis of the interview data: overall experience; expectations and importance of patient contribution; content of the discussion; composition of groups; facilitation of discussion; layout, including consideration of special needs; and timing.

In the following illustrative quotes, Consumer Forum members are coded C1–5 and researchers are coded R1–5, with 'a' and 'b' denoting the two researchers on each table. Patients with the same numbers as researchers were on the same table. More examples of participant responses for each section are in online supplemental appendix D.

### Experience and content
#### Overall experience (what did you get out of the session?)

Reflections on the overall experience were overwhelmingly positive. Both researchers and patients stated that the session had met or exceeded their expectations, that they learnt something from being part of it and that they would want to repeat it.

> All the topics were relevant, it was well planned, relevant to patients with cancer, living with and beyond cancer… I would like to see it repeated because I think it was a very fruitful event. (C3)

However, there were also some comments on difficulties or limitations of the session (see online supplemental appendix D, box 1).

### Expectations/importance of patient contribution

Both researchers and patients articulated the value of the contribution from members of the NCRI Consumer Forum. The patients and their carers felt that their input was valued, that they could improve the clarity of the research questions asked by the researchers, and help in considering the practicalities involved in the research.

> Working together towards improving outcomes for patients is the best way to achieve reliable results… As a colleague on an equal basis everyone working side-by-side to discuss the research and receiving equal consideration it's so important I think to provide this opportunity. (C3)

Although it was clear that input from the patients had made a concrete difference to the researchers' work, some patients felt that more needed to be done to show them the value of their input after the event.

> I don't know that I've heard very much what about what will actually happen, what the next steps are…  (C2)

The researchers' responses focused on the new insights and perspectives that the patients had brought, including how it helps to remind us that there are real people represented by each data point (online supplemental appendix D, box 2).

### Content of discussion

Inevitably, the content of the discussions varied greatly between groups. It was clear from the responses of both patients and researchers that some of the projects were less amenable to this format of discussion than others, for example, when the project was primarily about methodological development.

> There were a few steps that the Consumers needed to understand before they could really answer things that would be useful to me…all of these concepts were quite tricky to grasp. (R1a)

Where there was synergy, the discussion flowed easily, the time seemed too short and the suggestions made were fruitful and well received. At other tables, it took longer for mutual understanding to develop, although it was generally reached in the end (online supplemental appendix D, box 3).

### Organisation of groups

The final allocation saw six to eight patients at each table. Ultimately, 10–12 individuals participated in each group, including researchers, patients and, in some cases, other observers.

### Composition of groups

Researchers and patients were generally positive about the size and overall composition of the groups. The size of each group allowed for a variety of backgrounds to be

represented at each table, which was viewed as beneficial to the discussions.

It was clear, though, that there is a happy medium for the number of participants that provides both diversity of experience and sufficient time for all participants to contribute to fruitful discussion.

> I think it was fine. I wouldn't put more. But less is maybe too… you may not achieve what you want. I think you need the diversity, but not too many people. (R2a)

Due to the size and diversity of the groups in terms of experiences, both of cancer and its treatment, and in giving this kind of feedback to researchers, one researcher felt that time was spent at the start in order for everyone to '*get on the same page*' (R4a). However, researchers and patients both recognised that there was a lack of diversity and representation of some demographic groups, because members of the NCRI Consumer Forum inevitably represent a subset of the population, and perhaps especially as the majority of those who attended had been members for at least 2 years (online supplemental appendix D, box 4). NCRI Consumers are recruited in open competition, mainly to serve on academic research committees or other research groups working at a strategic level, and a successful applicant is required to have some prior knowledge or experience of cancer research.[20]

### Facilitation of discussion

The researcher presenting the topic at each table facilitated the discussions. Feedback from the patients and other researchers on how that role was performed was generally positive, but it was suggested that combining facilitation with explaining the research ideas and the main goals of the session was too much, and that an 'independent' facilitator might be helpful. There was also a recommendation that it might be an advantage to have more senior moderators, who would feel more comfortable being directive in keeping the discussions on topic.

> I think if maybe it had been a more senior person there…who is almost monitoring, keeping an eye on what's going on, giving a helping hand if somebody looks a little bit adrift, they could intervene at that point and get things back on track for them. (C5)

One point identified for aiding facilitation was a suggestion that more structure to the questions might have helped direct the discussions. Clarifying at the outset what the main point is, where contributions would help and which issues were in fact less important may have made the session more fruitful.

Given that the researcher at each table was both presenting the research and facilitating the discussions, having a separate note-taker was appreciated. Another suggestion was to encourage others to make notes, and that the inclusion of visual aids, such as whiteboards or flip charts, might help clarify the main issues, and focus

and direct the discussions (online supplemental appendix D, box 5).

### Layout and timing
#### Layout of the room
The impact of the layout of the room featured for both patients and researchers, who felt that the room was too small and the tables too close together. There was some concern that those with hearing difficulties were disadvantaged by the layout, and the patients, predominantly, felt that the discussions would have been better had the layout had been different.

> …there were loads of tables in the one room. It was actually quite difficult to hear so that was frustrating in some ways (C2)

Some also made recommendations about having each group in a separate room, although others felt the 'buzz' of the room helped create a good atmosphere (online supplemental appendix D, box 6).

#### Consideration of special needs
In their feedback, the patients gave some useful comments regarding consideration of the special needs of their group. In addition to the room layout impacting those with hearing impairments (as above), it was suggested that materials should have been provided for note-taking.

> Everyone having pen and paper to write things down, but having our own space to write things, a turn to speak and then write stuff down to be collected that might help. (C1)

They also suggested reducing the intensity and length of the day, especially considering the long-term health conditions of some of the patients, some of whom were still undergoing treatment (online supplemental appendix D, box 7).

#### Length of the session
The session was scheduled after the lunch break for 40 mins, followed by a 45 mins plenary session. Given the length of time it took for each table to settle and begin the discussions, and after a few minutes of introduction and background to the topic by the researcher, there was roughly 30 mins for open dialogue.

Many comments indicated that the time was too short, based partly on the numbers of patients at each table, but mainly on the complexity of the topics being presented.

> …in trying to put those objectives of my project in the most specific way and in lay terms, understandable way, I think by the time we bridge these initial difficulties we run out of time. (R4a)

In a few groups it seemed that the discussions only made progress towards the end of the time allocated. Both patients and researchers commented that a substantial part of the time was taken at the start of the session to develop a good understanding of the project and where

the researchers needed insight from the patients (online supplemental appendix D, box 8).

## Summary of postevent evaluation by patients

Two-thirds (64%, 29/45) of patients responded to the request for evaluation immediately following the event. The Dragons' Den session was rated as the most informative session of the day by patients. In fact, 86% rated the group session informative for their 'personal development as a consumer in cancer research' at some level, and almost three-quarters of attendees (72%) found the session useful for their role as an NCRI Consumer to some degree (online supplemental appendix B).

Overall, hosting of each table was overall considered effective ('very effective': 9/29, 31.0%), but just under half of respondents found it only 'slightly effective' (13/29, 45%), with a few patients rating it neutral or ineffective, or exceptional. Most respondents found the event well organised (93%) and the venue location good (90%), but fewer found the room comfortable (59%).

## RECOMMENDATIONS FOR THE FUTURE

The lessons learnt from this experience have helped inform the CSG's interactions with patients. We hope they will offer guidance to other researchers who want to arrange similar sessions. Although the epidemiological and methodological research in the CSG differs from that on which patients are more used to being consulted (eg, designing clinical trials), the experience can be fruitful for both patients and researchers.

### Suggestions for running a similar patient involvement session
#### Before the event
*Send detailed 'pre-work' which has been piloted with one or two patients first, ideally cowritten by them*
Explain each study succinctly. Outline the specific questions that the researchers wish to explore. Encourage patients to send questions in advance, to show the researchers what they might want to discuss or have clarified.

#### On the day
*Ensure enough space between tables so that all can be heard*
Ensure the layout is appropriate for the size of groups. Seating arrangements will be dependent on the available space, but ideally, the tables should be sufficiently well spaced that discussions at neighbouring tables are not a distraction.

While some groups may find background silence daunting and the buzz of conversation may help stimulate people to speak or make them feel more relaxed, for this group of experienced patients, most of whom were ready and willing to talk, a layout with more space would have been preferable. In another study that used a similar format of patient engagement, Hill *et al*[3] found that they successfully dealt with initial issues of noise by spacing the tables out at future events.

We would suggest only two to three groups (tables) per room if possible, so they can be spread as widely as possible. This allows a sense of privacy and ensures that everyone at each table can hear and be heard.

We would recommend excluding those who are not directly involved in the session from the room (eg, organisers, observers), as we found that these people add to the level of noise and can discourage openness.

#### Provide note-taking facilities and encourage their use
It is useful to provide pen and paper or post-it notes, and encourage notes to be written down that can be shared with the researchers during and after the session.

A whiteboard or flip chart for each group would enable diagrams and technical terms to be written up, so that suggestions can be recorded for all to see. It is a good idea to photograph the boards or flip charts at the end of the session, before they are taken down.

#### Allow plenty of time
If possible, take a whole day. Rather than having a one-off session, the day could be split into three discrete sessions, with breaks in between. Time can then be spent first on introductions, as well as allowing patients time to introduce themselves and share their experiences. These all add to researchers' understanding, even if they are not directly relevant to the questions that ultimately need to be answered about each project.

Second, allow a generous amount of time for the introduction and initial discussion of the topic. This session can focus on general questions the patients might have to clarify the work, or arising from their 'pre-work'. It should end with the researcher laying out the specific questions on which they would like to focus in the final discussion.

Lastly, run the Dragons' Den session, which should then focus on eliciting specific advice for the researchers to solve the problems they may have been having with study design, or on how patients' experiences could help improve their research question.

#### Have an independent facilitator for each group
While this type of activity is suitable for all researchers, no matter their seniority, career stage or experience of patient and public involvement (PPI), we recommend having an 'independent' facilitator for each group who is not directly involved in the research. This should be a senior researcher with experience of chairing meetings who can be diplomatic but firm, while keeping the discussions focused, ensuring everyone is encouraged to speak and summarising the main points to check for clarity.[3]

#### After the event
#### Follow-up with patients
As well as the usual evaluation forms, we recommend that researchers send follow-up notes to participants so that they can add further comments after having had a chance to reflect. This will add to the richness of experience gained from the day, and use the patients' knowledge to the fullest extent.

It is also important to inform the people who have contributed to the research about what has happened as a result of their input and to let them know how the research has changed or developed as a result. This may be in the form of asking them to comment on research proposals based on the project discussed or in a brief report on its progress.

### Remaining challenges

Several challenges remain. The composition of patient groups was not fully representative of the population of patients with cancer. This was also noted in a systematic review of PPI in cancer research,[10] which found that PPI participants tend 'to be well-educated female participants from ethnic majority groups' (p 14). Harder-to-reach groups should be invited, if possible, including more disadvantaged patients or their carers, those of different ethnicities and those with language barriers, or their representatives. By definition, it is difficult to include these groups, but efforts to approach them should be considered. For example, if a session were focused on specific cancers (or other disease areas) one or two charities focused on those conditions might be asked to invite participants.

We would also like to be able to include patients with more special needs. Consideration should be given to the support such patients would need, and how it can best be provided.

The CSG is fortunate to have access to numerous teaching rooms, but space was still an issue. In some locations, space may be even more limited. We would still suggest attempting a Dragons' Den event, but perhaps for one or two groups at a time.

### CONCLUSION

We have shared the lessons from one research group's experience of engaging with patients and their carers, in the hope of encouraging other researchers to carry out this kind of consultation, while starting from a more informed position.

The Dragons' Den process is extremely valuable to researchers engaged in all kinds of research, including epidemiological research based on routinely collected data. As well as receiving constructive advice on current research projects, it was also a chance for us to reconnect with the reason we do this research.

From the patients' perspective, participants were keen to contribute to work that would inform research and change policy. They emphasise the need for action and for everyone to 'raise their voices' to work for easier access to data for epidemiological research, and to ensure that patients remain at the heart of all the research we do.

This event was held well before the COVID-19 pandemic. For now, the way we run events and activities must change, especially if those events would include people in vulnerable groups, for example, those undergoing treatment for cancer. We believe that very successful

patient involvement sessions can still be run using online platforms, using many of the suggestions we have made above.

Similar considerations of prework, group size and composition, facilitation, note-taking and timing will be useful in planning online events, and some will come to the fore even more. For instance, we believe that an experienced, independent facilitator will be particularly important in this setting.

Although online events can feel very different, we would argue that most of the same considerations are vital for researchers hoping to run such an event. Having to work remotely should not prevent us from engaging with and involving the patients and future patients whom we hope will benefit from our research.

**Acknowledgements** We gratefully acknowledge all the participants of the Dragons' Den session, in particular those who were interviewed after the event. We also acknowledge Dr Edmund Njeru Njagi for his contribution to the collation of feedback after the event and comments on the initial paper outline. Finally, we acknowledge Simon Callaghan who provided the photos from the event and all those who participated and allowed their images to be used. All participants were informed about photography before the event and on the day, and an option to opt out of appearing in any published images was provided and respected.

**Contributors** All authors conceived and planned the study. MM conducted and transcribed the interviews, gathered and analysed the data. MM and YA summarised the data, reviewed the literature and wrote the manuscript. RS, MC and BR provided comments on the manuscript and all authors have seen and approved the final version of the manuscript for publication.

**Funding** YA is supported by Cancer Research UK (grant number C7923/A18525).

**Competing interests** None declared.

**Patient and public involvement statement** One of the coauthors (RS) is a patient representative and former chair of the NCRI Consumer Forum of patients with cancer and carers. The project idea was conceived together with RS and he was involved in the study concept and design.

**Patient consent for publication** Not required.

**Ethics approval** This study was approved by the LSHTM Ethics Committee (ref 14502).

**Provenance and peer review** Not commissioned; externally peer reviewed.

**ORCID iDs**
Yuki Alencar http://orcid.org/0000-0001-9913-6423
Bernard Rachet http://orcid.org/0000-0001-5837-7773

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
