## [Reviewer comments · BMJ Open]

ARTICLE DETAILS

TITLE (PROVISIONAL)	Fleshing out the data - when epidemiological researchers engage with cancer patients and carers: learning lessons from a patient involvement activity
AUTHORS	Morris, Melanie; Alencar, Yuki; Rachet, Bernard; Stephens, Richard; Coleman, Michel

VERSION 1 – REVIEW

REVIEWER	Elizabeth Davies King's College London, UK
REVIEW RETURNED	25-Mar-2020

GENERAL COMMENTS	Thank you. This is a well-written paper which presents a thoughtful exploration of the experience of bringing patients and their carers together with cancer epidemiologists to define new questions for research projects. The authors present their experience and lessons learnt from using a specific 'Dragon's Den' approach to defining new studies. I think this paper presents enough detail for other groups to follow practically and learn from, and also represents a radical move forward for the field. While some cancer epidemiologists do work closely with patients and carers to define their questions, not all do and this will hopefully increase confidence and collaboration in the field.
--

REVIEWER	W.H van Harten Netherlands Cancer Institute
REVIEW RETURNED	29-Mar-2020

GENERAL COMMENTS	This paper is a report of an activity on involving patients in providing insight, advice and ideas on planned research, for which a meeting with cancer survivors was used. The report reads as a factual description of activities, patient input and researchers' experiences. One would expect a more thorough literature review as a start of this type of project so a framework or format for evaluation was drawn before hand and described in the methods. As this is missing, no format for set-up, no theoretical backup and instruction for chairs and participants nor a structure or semi structure for the way of presenting the research ideas and discussion is provided. We can thus not verify whether every group was presented with research ideas the same degree of development (that might evoke different responses/input in different stages); whether they followed the same discussion steps etc, etc. The results seem to vary from very practical o self evident: to many groups in a small space is not a good idea, a lot of noise is
---

	not suitable for people with hearing impairments, an independent discussion leader is better etc, etc. The title/introduction claims the research to deal with epidemiological research and secondary data analysis. This topic is not elaborated in any structural way. Further cancer survivors were mentioned as participants but nothing is further done with their backgrounds nor a specification on the degree of their experience in providing this type of feedback. Lastly no information is given on the researchers/discussion leaders and their skills, reputation as researchers or otherwise and experience in this field of patient involvement. The boxes with discussion summaries do not read easily as a structure is missing and this cannot be related to the paper and it is not clear why quotes appear in the primary text or why just in the box (and what is left out), In all it seems merely a report of a discussion meeting, without clarity on the scientific structure and value, and due to lack of theoretical foundation and structure it is not clear what this adds to the theory and knowledge on involving patients and patient advocates in conceiving research and/or decisions on research design, apart from some practical, sometimes common sense type of suggestions.
--	--

REVIEWER	Mia Bierbaum Macquarie University, Australia
REVIEW RETURNED	03-Apr-2020

GENERAL COMMENTS	Summary  • This project is a meaningful and important piece of research. It has been concisely presented and provides useful recommendations for researchers intending to include patient involvement in future quantitative and epidemiological research projects. I have several minor suggestions for the manuscript. Abstract  • Clear and concise. Strengths and limitations  • Expand PPI the first time it is used (second to last dot point). Introduction  • “They highlight the wider consideration of making health research more transparent” reads slightly awkwardly and could be rephrased. P5, line 27 • Elaborate on what the “plenary presentations and discussion” involved and how they allowed you to capture the conversation and recurrent themes that emerged. This sounds like a fundamental component of the research project, that deserves additional explanation. P6, Line 3 • Include references for the methods used e.g. timing of Dragon’s den sessions, p6, line 21 and thematic analysis, P6, line 34 Summary of reflections and feedback  • Briefly explain how the presentation of findings relates to the thematic analysis method used. (e.g. the identification of x number of themes and subthemes; experiences, organisation of groups, layout and timing, etc as appropriate). • Briefly describe who conducted the thematic analysis, including any triangulation, team coding or use of coding software. See The
--

	Standards for reporting Qualitative Research https://www.equator-network.org/reporting-guidelines/srqr/ Recommendations for the future, conclusion • These sections were clear and summarised the findings of the study well.
REVIEWER	Mei Krishnasamy University of Melbourne and the Peter MacCallum Cancer Centre, Melbourne, Victoria, Australia
REVIEW RETURNED	05-Apr-2020
GENERAL COMMENTS	This is an interesting communication piece and the lessons shared will be of value to others wishing to learn how to create opportunity to better engage the public and patients in their research. The presentation of a set of instructions or guidance about how to go about setting up a similar event is a helpful aspect of the paper and I am sure will be useful and used by readers looking for advice to strengthen this area of their work. I have two minor recommendations: 1) for those unfamiliar with the concept of a "Dragon's Den" it may be helpful to explain this, and 2) there is no mention of consent to publish photos of participants. Please can there be reference to peoples' consent for this.

VERSION 1 – AUTHOR RESPONSE

Reviewer: 1

Please leave your comments for the authors below

Thank you. This is a well-written paper which presents a thoughtful exploration of the experience of bringing patients and their carers together with cancer epidemiologists to define new questions for research projects. The authors present their experience and lessons learnt from using a specific 'Dragon's Den' approach to defining new studies. I think this paper presents enough detail for other groups to follow practically and learn from, and also represents a radical move forward for the field. While some cancer epidemiologists do work closely with patients and carers to define their questions, not all do and this will hopefully increase confidence and collaboration in the field.

Thank you for your positive comments.

Reviewer: 2

Please leave your comments for the authors below

This paper is a report of an activity on involving patients in providing insight, advice and ideas on planned research, for which a meeting with cancer survivors was used. The report reads as a factual description of activities, patient input and researchers' experiences.

Thank you for this accurate summary.

One would expect a more thorough literature review as a start of this type of project so a framework or format for evaluation was drawn before hand and described in the methods.

We would have done such a literature review without any doubt if this article was about a research project. This was however not the case and was rather a patient involvement activity which was summarised and the findings set out as a communication piece to share with other researchers. The primary purpose of the event was to involve patients in our ongoing research, as such it was not planned with a project framework or format for evaluation in mind.

We have added wording at the end of the introduction which we hope will emphasise the purpose of the piece to the readers.

As this is missing, no format for set-up, no theoretical backup and instruction for chairs and participants nor a structure or semi structure for the way of presenting the research ideas and discussion is provided. We can thus not verify whether every group was presented with research ideas the same degree of development (that might evoke different responses/input in different stages); whether they followed the same discussion steps etc, etc.

For the purpose of the event, and the recommendations in the communication piece, the stage of development of the research ideas was not relevant. In fact, the value of this event can be for research ideas at various stages, and the discussions steps can be led by the researcher and research topic being presented. The aim was not to compare between the different groups but to present what worked overall and what could be improved.

The results seem to vary from very practical o self evident: to many groups in a small space is not a good idea, a lot of noise is not suitable for people with hearing impairments, an independent discussion leader is better etc, etc.

Indeed, we were aiming for practical advice, a list of helpful ideas that would benefit others planning a similar event.

The title/introduction claims the research to deal with epidemiological research and secondary data analysis. This topic is not elaborated in any structural way.

The purpose here was to make the distinction between the involvement of patients (i) in interventional studies, which lends itself more obviously to PPI, and (ii) in epidemiological/secondary data analysis research, in order to encourage those who do that type of research to consider innovative options for PPI. The details of the topic, beyond this broad description, is not necessary for the purpose of the communication piece.

Further cancer survivors were mentioned as participants but nothing is further done with their backgrounds nor a specification on the degree of their experience in providing this type of feedback.

This information was not collected, but we acknowledge the important concerns about diversity, experience etc. Indeed, we do comment on the lack of diversity (see Composition of groups) and the need for broader representation in such activities (see Remaining challenges). We have added some wording (on p.6 of tracked document, in Composition of Groups) to acknowledge that participants had varying experience of providing this kind of feedback as members of the NCRI Consumer Forum.

Lastly no information is given on the researchers/discussion leaders and their skills, reputation as researchers or otherwise and experience in this field of patient involvement.

Such information is not key in the narrative of this patient involvement, but we do nevertheless comment on the potential benefit of involving more senior researchers as facilitators (see Have an independent facilitator for each group in Suggestions) and have added a line about the suitability of this type of activity for those with any level of experience (of research or PPI) in that section.

The boxes with discussion summaries do not read easily as a structure is missing and this cannot be related to the paper and it is not clear why quotes appear in the primary text or why just in the box (and what is left out),

Word count will not allow all comments to be included in primary text so the most relevant quotes

were selected for this purpose. The boxes have been provided as supplementary materials and are now referred to again in the main text. As part of the submission process we were asked to remove reference to the correlating box numbers from the main text. We would be happy for the editors to decide how best to present/link these together to make the structure, and how it relates to the paper, clearer.

In all it seems merely a report of a discussion meeting, without clarity on the scientific structure and value, and due to lack of theoretical foundation and structure it is not clear what this adds to the theory and knowledge on involving patients and patient advocates in conceiving research and/or decisions on research design, apart from some practical, sometimes common sense type of suggestions.

The purpose of the communication piece was to share the experience and provide practical suggestions, what may seem common sense to one person may not be as evident to others. This reviewer's comment is in contrast to the other reviewers who found the piece detailed enough and helpful.

Reviewer: 3

Please leave your comments for the authors below

Summary

- This project is a meaningful and important piece of research. It has been concisely presented and provides useful recommendations for researchers intending to include patient involvement in future quantitative and epidemiological research projects. I have several minor suggestions for the manuscript.

Thank you for your positive comments and suggestions.

Abstract

- Clear and concise.

Strengths and limitations

- Expand PPI the first time it is used (second to last dot point).

PPI has been expanded in the first line of the abstract and the order of the bullet points changed.

Introduction

- "They highlight the wider consideration of making health research more transparent" reads slightly awkwardly and could be rephrased. P5, line 27

This has been amended to "They also highlight importance of making health research more transparent and publicly accountable"

- Elaborate on what the "plenary presentations and discussion" involved and how they allowed you to capture the conversation and recurrent themes that emerged. This sounds like a fundamental component of the research project, that deserves additional explanation. P6, Line 3

The purpose of the "plenary presentations and discussion" was to provide a wrap up of the session and feedback to the whole group the main points of discussion on each table. Although some post Dragons' Den feedback does normally take place at the NCRI events, it is not a fundamental component and does not always take place (depending on the time available). Although it was a valuable and appreciated part of the day, it did not add to themes developed in this paper, however, which emerged from the subsequent interviews with selected participants which focused on the Dragons' Den session itself. The paper has been amended to clarify the function of this part of the event (see An example of cancer patient involvement).

- Include references for the methods used e.g. timing of Dragon's den sessions, p6, line 21 and thematic analysis, P6, line 34

The typical timing of the sessions was advised to us by the patient representatives who helped us plan the day. The methodology of the NCRI Consumer Forum Dragons' Den is as yet unpublished, but the Dens were started in 2012 and had a proven functional design at the time of our meeting which we used.

Some more detail on the thematic analysis and two references have been added on p.4 (of the tracked document).

Summary of reflections and feedback

- Briefly explain how the presentation of findings relates to the thematic analysis method used. (e.g. the identification of x number of themes and subthemes; experiences, organisation of groups, layout and timing, etc as appropriate).

This has been added at the beginning of the Summary of reflections and feedback section

- Briefly describe who conducted the thematic analysis, including any triangulation, team coding or use of coding software. See The Standards for reporting Qualitative Research <https://www.equator-network.org/reporting-guidelines/srqr/>

This detail was added in the methods just above the Summary of reflections and feedback

Recommendations for the future, conclusion

- These sections were clear and summarised the findings of the study well.

Reviewer: 4

Please leave your comments for the authors below

This is an interesting communication piece and the lessons shared will be of value to others wishing to learn how to create opportunity to better engage the public and patients in their research. The presentation of a set of instructions or guidance about how to go about setting up a similar event is a helpful aspect of the paper and I am sure will be useful and used by readers looking for advice to strengthen this area of their work.

Thank you for your positive comments and suggestions.

I have two minor recommendations: 1) for those unfamiliar with the concept of a "Dragon's Den" it may be helpful to explain this,

Some wording has been added in the Introduction where "Dragons' Den" is first mentioned.

and 2) there is no mention of consent to publish photos of participants. Please can there be reference to peoples' consent for this.

Information on this was provided to BMJOpen as part of the submission process. A statement on informed consent has been added to the acknowledgments.